# A disproportionality analysis of FDA adverse event reporting system (FAERS) events for methimazole and propylthiouracil

Yun Li[1][☺], Hang Li[2][☺], Qiong Sun[3], Qin Long[iD][3]*

1 Department of pharmacy, Yueyang Central Hospital, Yueyang, Hunan, China, 2 Department of Pharmacy, The Second Xiangya Hospital, Central South University, Changsha, Hunan, China, 3 Department of Endocrinology, Yueyang Central Hospital, Yueyang, Hunan, China

☺ These authors contributed equally to this work.
* 595278228@qq.com

## Abstract

### Background

Methimazole and propylthiouracil are the most common antithyroid drugs. We assessed the safety signals associated with methimazole and propylthiouracil by data mining the FDA pharmacovigilance database.

### Methods

Data were retrieved from the FAERS database from the 1st quarter of 2004 through the 4th quarter of 2023. A disproportionate analysis of reporting advantage ratios was used to assess potential associations between adverse events and methimazole/propylthiouracil.

### Results

A total of 17,379,609 reports were extracted, of which 5,317 cases of methimazole and 1,761 cases of propylthiouracil were classified as primary suspect reports. After combining the same primary ID, 1586 patients with methimazole and 446 patients with propylthiouracil were retained. We observed 8 categories of SOCs with a reported number ≥ 30 for methimazole and 12 categories of SOCs with a reported number ≥ 10 for propylthiouracil. The median time to adverse events in patients with methimazole was 31 days, with an interquartile range of 31–74 days. The median time to adverse events in patients with propylthiouracil was 90 days, with an inter-quartile range was 20–388.5 days.

**Data availability statement:** All relevant data are within the paper and its Supporting Information files.

**Funding:** This work was supported by the Yueyang Central Hospital (No. YYSZXYN2023019 to Y.L.).

**Competing interests:** The authors have declared that no competing interests exist.

## Conclusion

Our study provided a more in-depth and extensive understanding of adverse events that may be associated with methimazole and propylthiouracil, which will help to reduce the risk of adverse events in the clinical treatment of methimazole and propylthiouracil.

## 1. Introduction

Thyrotoxicosis is the clinical manifestation of excess thyroid hormone action at the tissue level. Thyrotoxicosis be caused by an increase in synthesis and secretion of thyroid hormones by the thyroid or the release of stored thyroid hormone from the gland or extrathyroidal sources of thyroid hormone. In clinical practice, Graves' disease, toxic nodular goiter, and thyroiditis are the most common conditions. The global epidemiology of thyroid disease is strongly related to population iodine status. [1,2] In iodine-sufficient areas, the incidence of thyrotoxicosis is about 50 cases per 100,000 people per year. [3,4] In iodine-deficient areas, the prevalence of hyperthyroidism is higher: up to 10–15%. [5,6]

Antithyroid drugs (ATDs) are now an increasingly popular treatment option for patients with thyrotoxicosis in many countries. [7] Methimazole (MMI) and propylthiouracil (PTU) are the drugs of choice for thyrotoxicosis, and they inhibit the thyroid hormone synthesis by blocking the thyroid peroxidase-mediated iodination of tyrosine residues in thyroglobulin. [8] Minor side-effects occur in about 5% of patients using ATDs, including pruritus and gastrointestinal distress. However, some major adverse events (AEs) may have considerable impacts, such as agranulocytosis, hepatotoxicity, and vasculitis, and they are feared reaction. [7] Although agranulocytosis reportedly occurs in less than 0.5% of patients receiving ATDs, it can be fatal and should be prevented. Agranulocytosis usually occurs during the first 3 months of treatment with ATDs and may be manifested as fever, sore throat, and diarrhea. [9] Patients should be alerted for the occurrence of these symptoms. If agranulocytosis is confirmed, ATDs should be discontinued permanently. The prevalence of severe hepatotoxicity was 2.5%, and the incidence of severe hepatotoxicity is significantly higher in PTU than with MMI. [10] A number of studies have also demonstrated that ATDs cause antineutrophil cytoplasmic antibody-associated vasculitis (AAV), with clinical manifestations of fever, malaise and cutaneous vasculitis. [11–13] AAV is associated with high morbidity and mortality and requires appropriate interventions. The administration of MMI may also increase the risk of acute pancreatitis in patients, but there is disagreement about this effect. [14,15]

There is a difference in the incidence of side effects between MMI and PTU, and there are limited comprehensive analyses and comparative studies of the various AEs caused by ATDs. Studies have been conducted using the US FDA Adverse Event Reporting System (FAERS) to explore various AEs associated with a number of other drugs, providing new insights into the monitoring, surveillance and

management of adverse drug reactions. [16–22] There is a lack of studies using FAERS to examine AEs of MMI and PTU. Therefore, this study will examine AEs of MMI and PTUs using FAERS to gain further insight into adverse events associated with the use of ATDs.

## 2. Methods

### 2.1. Study design and data source

FAERS, the United States' passive surveillance system for adverse drug events, is one of the largest pharmacovigilance repositories, with virtually worldwide population coverage. We conducted an observational, retrospective disproportionate analysis of MMI and PTU based on FAERS data [23]. Based on when the FDA approved MMI and PTU for marketing, we searched for all case reports of major suspected drug use of MMI and PTU reported to FAERS between the first quarter of 2004 and the fourth quarter of 2023.

### 2.2. Data extraction and descriptive analysis

Raw data downloaded from the official FAERS database website include: patient demographic information (DEMO), drug information (DRUG), adverse event information (REAC), patient outcome information (OUTC), report source information (RPSR), drug therapy date information (THER), and drug indication (INDI). The seven raw data were imported into the R Programming Language (v4.3.3; USA) and deduplicated. The primary ID is the report number, a unique identifier that can be associated in individual tables. Case ID identified the number of FAERS cases, therefore Case ID was chosen as a filter in our study to remove duplicate records. We selected role_cod as the primary suspect (PS) in the drug file. We used R language for deduplication, identifying duplicates based on primary ID and qualifying MMI/PTU as the PS medication AE in the code. AEs in FAERS are coded using the preferred PT in the standardized Medical Dictionary of Regulatory Activities (MedDRA), which contains 27 SOCs. Additionally, a PT can be linked to multiple SOCs in MedDRA. Therefore, we used MedDRA 26.0 to categorize the AEs in each report to derive the corresponding SOC level. Our study included all PTs below SOC in the FAERS database. The multistep flowchart for data extraction, processing, and analysis is shown in Fig 1.

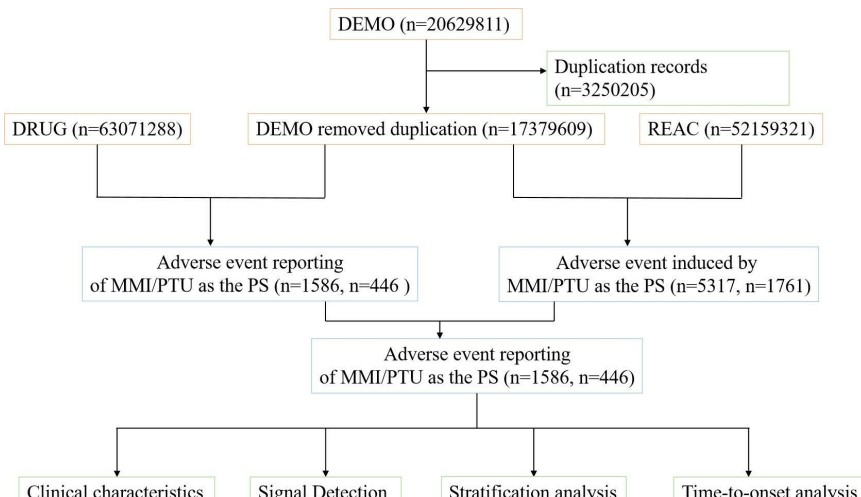

**Fig 1. The process of selecting methimazole and propylthiouracil related adverse events from FAERS database.** MMI, methimazole; PTU, propylthiouracil.

## 2.3. Statistical analysis

We used disproportionality analysis to detect spontaneous signals. In assessing potential associations between MMI/PTU and AEs, the ratio-of-reporting (ROR), the proportional reporting ratio (PRR), the empirical Bayes geometric mean (EBGM) and Information Component (IC) were used separately. AE signals with at least 20 records were analysed and signals that simultaneously met the criteria of the four algorithms with a lower limit of 95% confidence interval (CI) > 1.0 were selected.

## 3. Results

### 3.1. Descriptive analysis

After excluding duplicates, a total of 17,379,609 AE reports were obtained from the FAERS database, including 5,317 AEs associated with MMI in 1,586 patients and 1,761 AEs associated with PTU in 446 patients (Fig 1). Detailed clinical characteristics are summarized in Table 1. Of the MMI reporting patients by gender, 1,031 (65.1%) were female and 299 (18.9%) were male. Age data were available for 1,060 patients, 935 of whom were older than 18 years. Of the 308 patients with weight data who reported MMI treatment, 68 were lower weight patients (<50 kg) and 21 were higher weight patients

**Table 1. Clinical characteristics of patients with MMI/PTU-associated adverse events.**

| Clinical characteristics | MMI | | PTU | |
|---|---|---|---|---|
| | Overall reports | Percentage | Overall reports | Percentage |
| **Sex** | | | | |
| Female | 1031 | 65.1% | 332 | 74.4% |
| Male | 299 | 18.9% | 69 | 15.5% |
| Missing | 254 | 16.0% | 45 | 10.1% |
| **Weight** | | | | |
| <50 kg | 68 | 4.3% | 19 | 4.3% |
| 50–100 kg | 219 | 13.8% | 43 | 9.6% |
| >100 kg | 21 | 1.3% | 3 | 0.7% |
| Missing | 1276 | 80.6% | 381 | 85.4% |
| **Age** | | | | |
| <18 years | 125 | 7.9% | 29 | 6.5% |
| 18-64.9 years | 730 | 46.1% | 273 | 61.2% |
| 65-85 years | 190 | 12.0% | 42 | 9.4% |
| >85 years | 15 | 0.9% | 1 | 0.2% |
| Missing | 524 | 33.1% | 101 | 22.6% |
| **Reporters** | | | | |
| Consumers | 374 | 23.6% | 32 | 7.2% |
| Health professional | 237 | 15.0% | 120 | 26.9% |
| Lawyer | 3 | 0.2% | 1 | 0.2% |
| Medical doctor | 342 | 21.6% | 143 | 32.1% |
| Other health-professional | 445 | 28.1% | 77 | 17.3% |
| Pharmacist | 99 | 6.3% | 19 | 74.4% |
| Registered nurses | 1 | 0.1% | / | / |
| Missing | 83 | 5.2% | 54 | 12.1% |
| **Reported country** | | | | |
| US | 1014 | 51.0% | 180 | 28.3% |
| Non-US | 540 | 47.1% | 254 | 69.0% |
| Missing | 30 | 1.9% | 12 | 2.7% |

(>100 kg). In addition, healthcare professionals and consumers reported 1,124 (71.1%) and 377 (23.8%) MMI-associated drug-related AEs, respectively. 1,546 overall AEs were reported for serious outcomes (Table 2), of which other serious outcome 725 (37.1%), hospitalization 550 (28.2%), and life-threatening 93 (4.8%). Among the patients of gender reported in PTU, 332 (74.40%) were females and 69 (15.50%) were males. Of the 65 patients with weight data who reported PTU treatment, 3 were higher weight patients (>100 kg) and 19 were lower weight patients (<50 kg). Age data were available for 345 patients, of which 316 (70.8%) patients were older than 18 years. In addition, healthcare professionals and consumers reported 359 (80.6%) and 33 (7.4%). PTU-associated medication-related AEs, respectively. 577 overall AEs were reported for serious outcomes (Table 2), with other serious outcome 326 (41.0%), hospitalization 186 (29.8%), and life-threatening 47 (7.5%). In addition to this, we have tabulated the percentage of serious incidents in the top five countries in terms of the number of reports using MMI/PTU (Fig 2)

### 3.2. Disproportionality analysis

Our analyses excluded AEs that were manifestations of the target disease pathology. Compared with non-MMI-related AEs, MMI-related AEs showed an imbalance in 35 AEs were reported in FAERS database in at least 20 cases and the lower limit of 95% confidence interval >1. Table 3 presented the results of the disproportionality analysis of MMI-related AEs. The results include 10 SOCs: 5 cases of PT were associated with blood and lymphatic system disorders (n = 330). 1 PT was associated with general disorders and administration site conditions (n = 662). 2 cases of PT were associated with injury, poisoning and procedural complications (n = 118). 1 PTs were associated with skin and subcutaneous tissue disorders (n = 20). 2 cases of PT were associated with musculoskeletal and connective tissue disorders (n = 56). 1 cases of PT were associated with immune system disorders (n = 26). 5 case of PT was associated with hepatobiliary disorders (n = 129). 2 cases of PT were associated with respiratory, thoracic and mediastinal disorders (n = 69). 3 cases of PT were associated with congenital, familial and genetic disorders (n = 64). 1 PT was associated with pregnancy, puerperium and perinatal conditions (n = 29). In order to gain a deeper understanding of the AEs of MMI/PTU by age, we performed an age-stratified cross-sectional analysis comparing the distribution of the number of the top five systemic diseases in terms of MMI/PTU across age groups (Fig 3). The top five systemic diseases in MMI are general disorders and administration site conditions (n = 485), blood and lymphatic system disorders (n = 379), skin and subcutaneous tissue disorders (n = 281), gastrointestinal disorders (n = 244), musculoskeletal and connective tissue disorders (n = 233). The results suggest that 18–64.9 years of age reported the most AEs among these five diseases; In PTU are general disorders and administration site conditions (n = 122), injury, poisoning and procedural complications (n = 70), skin and subcutaneous tissue disorders (n = 101), hepatobiliary disorders (n = 111), gastrointestinal disorders (n = 101). Same as the MMI that 18–64.9 years of age reported the most AEs among these five diseases.

**Table 2. MMI/PTU serious outcomes of overall adverse events reports.**

| Frequency statistics for outcome indicators | MMI | | PTU | |
|---|---|---|---|---|
| | Overall reports | Percentage | Overall reports | Percentage |
| Congenital abnormality | 79 | 4.0% | 38 | 6.1% |
| Death | 78 | 4.0% | 32 | 5.1% |
| Disability | 25 | 1.3% | 3 | 0.5% |
| Hospitalization | 550 | 28.2% | 186 | 29.8% |
| Life-threatening | 93 | 4.8% | 47 | 7.5% |
| Other serious outcome | 725 | 37.1% | 256 | 41.0% |
| Required intervention to prevent permanent impairment | 16 | 0.8% | 15 | 2.4% |
| Missing | 386 | 19.8% | 48 | 7.7% |

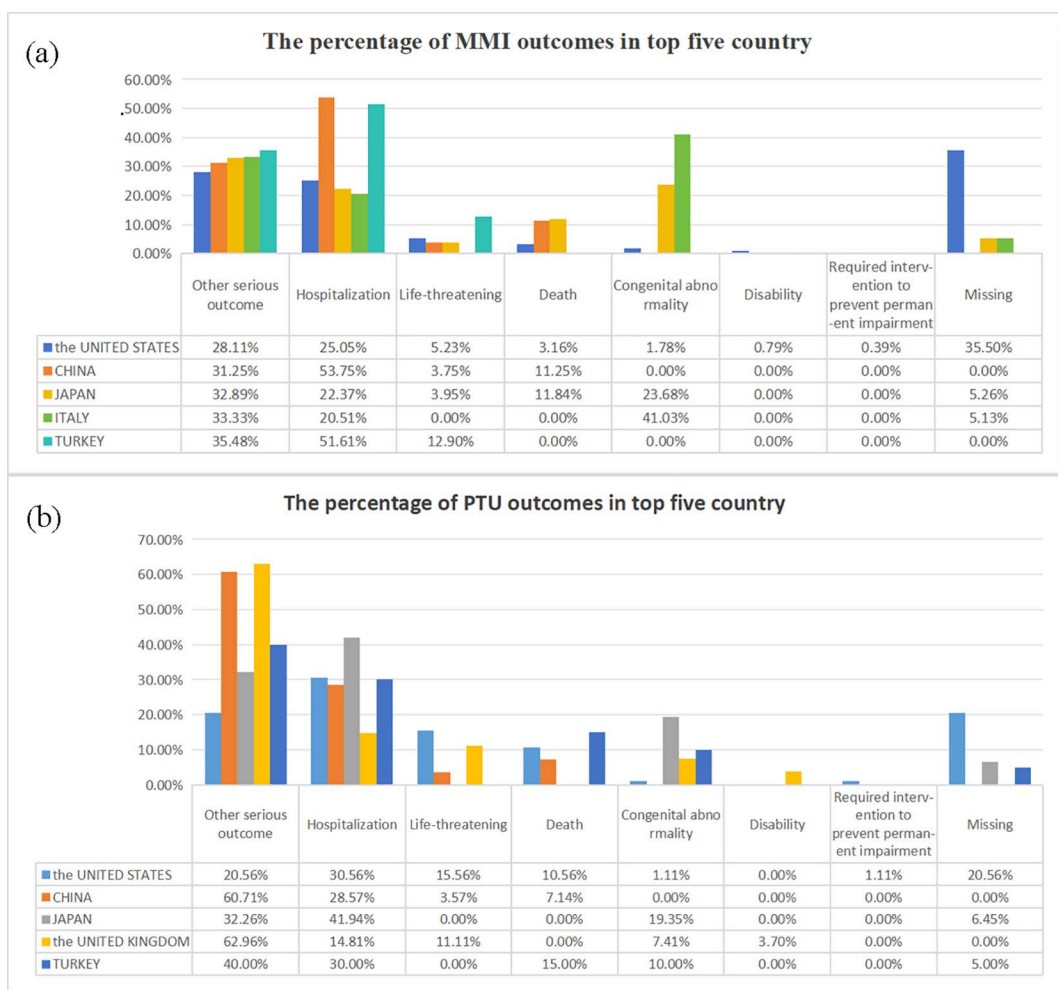

**Fig 2. The percentage of methimazole and propylthiouracil outcomes in top five country.** MMI, methimazole; PTU, propylthiouracil.

Compared with non-PTU-related AEs, PTU-related AEs showed an imbalance in 20 AEs were reported in FAERS database in at least 10 cases and the lower limit of 95% confidence interval >1. Table 4 presented the results of the disproportionate analysis of PTU-related AEs. The results include 12 SOCs: 1 PT was associated with immune system disorders (n = 62). 2 cases of PT were associated with injury, poisoning and procedural complications (n = 77). 1 case of PT was associated with blood and lymphatic system disorders (n = 28). 1 PT was associated with vascular disorders (n = 23). 2 cases of PT were associated with hepatobiliary disorders (n = 35). 1 case of PT was associated with respiratory, thoracic and mediastinal disorders (n = 26). 1 case of PT was associated with skin and subcutaneous tissue disorders (n = 13). 1 PT was associated with renal and urinary disorders (n = 10). 1 case of PT were associated with nervous system disorders (n = 12). 1 PT was associated with general disorders and administration site conditions (n = 11). 2 cases of PT were associated with investigations (n = 30). 1 PT was associated with surgical and medical procedures (n = 17).

### 3.3. MMI/PTU associated with acute SOC

The relationship between MMI/PTU and suspected SOC was analyzed using volcano plots (Fig 4). In each plot, the x-axis indicates the logarithm of the ROR. A positive x-axis indicates a greater incidence of drug-related SOC than other SOC.

**Table 3. ROR with 95% CI for all positive MMI-related AEs.**

| AEs | MMI.n (n = 5,317) | non.MMI.n (n = 52,154,004) | ROR (95%CL) | PRR (χ2) | EBGM(EBGM05) | IC(IC025) |
|---|---|---|---|---|---|---|
| **Blood And Lymphatic System Disorders** | | | | | | |
| Agranulocytosis | 192 | 14480 | 134.9 (116.69–155.95) | 130.06 (24274.61) | 128.37 (113.71) | 7 (5.34) |
| Neutropenia | 56 | 111792 | 3.31 (2.71–4.03) | 4.91 (174.85) | 4.91 (3.94) | 2.3 (0.63) |
| Febrile Neutropenia | 35 | 53966 | 8.1 (6.39–10.28) | 6.36 (158.22) | 6.36 (4.81) | 2.67 (1) |
| Pancytopenia | 24 | 46101 | 2.1 (1.64–2.67) | 5.11 (79.28) | 5.1 (3.65) | 2.35 (0.69) |
| Leukopenia | 23 | 42093 | 1.68 (1.3–2.17) | 5.36 (81.59) | 5.36 (3.8) | 2.42 (0.76) |
| **Congenital, Familial And Genetic Disorders** | | | | | | |
| Choanal Atresia | 22 | 97 | 1.5 (1.16–1.94) | 2224.7 (39859.32) | 1813.6 (1230.45) | 10.82 (9.14) |
| Exomphalos | 21 | 476 | 4.96 (3.81–6.45) | 432.75 (8663.59) | 414.5 (287.35) | 8.7 (7.02) |
| Dysmorphism | 21 | 2474 | 6.98 (5.27–9.25) | 83.26 (1692.44) | 82.57 (57.6) | 6.37 (4.7) |
| **General Disorders And Administration Site Conditions** | | | | | | |
| Pyrexia | 99 | 297539 | 2.78 (2.09–3.69) | 3.26 (156.28) | 3.26 (2.76) | 1.71 (0.04) |
| **Hepatobiliary Disorders** | | | | | | |
| Drug-Induced Liver Injury | 33 | 21785 | 2.82 (2.06–3.87) | 14.86 (426.09) | 14.84 (11.14) | 3.89 (2.22) |
| Jaundice | 26 | 23974 | 4.78 (3.47–6.57) | 10.64 (226.89) | 10.63 (7.7) | 3.41 (1.74) |
| Cholestasis | 25 | 15728 | 2.51 (1.81–3.47) | 15.59 (340.95) | 15.57 (11.2) | 3.96 (2.29) |
| Jaundice Cholestatic | 24 | 3101 | 6.4 (4.59–8.92) | 75.92 (1760.77) | 75.34 (53.8) | 6.24 (4.57) |
| Hepatotoxicity | 21 | 18114 | 81.32 (57.97–114.1) | 11.37 (198.49) | 11.36 (7.93) | 3.51 (1.84) |
| **Immune System Disorders** | | | | | | |
| Anti-Neutrophil Cytoplasmic Antibody Positive Vasculitis | 26 | 1653 | 1.94 (1.36–2.77) | 154.28 (3898.37) | 151.91 (109.77) | 7.25 (5.58) |
| **Injury, Poisoning And Procedural Complications** | | | | | | |
| Exposure During Pregnancy | 69 | 84473 | 1.67 (1.16–2.41) | 8.01 (423.8) | 8.01 (6.56) | 3 (1.33) |
| Foetal Exposure During Pregnancy | 49 | 69387 | 155.04 (105.14–228.61) | 6.93 (248.65) | 6.92 (5.47) | 2.79 (1.12) |
| **Musculoskeletal And Connective Tissue Disorders** | | | | | | |
| Polyarthritis | 34 | 4127 | 5.38 (3.57–8.1) | 80.81 (2658.28) | 80.16 (60.38) | 6.32 (4.66) |
| Rhabdomyolysis | 22 | 35214 | 2233.94 (1405.15–3551.58) | 6.13 (94.41) | 6.12 (4.31) | 2.61 (0.95) |
| **Pregnancy, Puerperium And Perinatal Conditions** | | | | | | |
| Premature Baby | 29 | 26761 | 6.15 (4.05–9.35) | 10.63 (252.84) | 10.62 (7.82) | 3.41 (1.74) |
| **Respiratory, Thoracic And Mediastinal Disorders** | | | | | | |
| Oropharyngeal Pain | 38 | 78463 | 434.46 (280.4–673.15) | 4.75 (112.63) | 4.75 (3.64) | 2.25 (0.58) |
| Pleural Effusion | 31 | 52711 | 83.59 (54.35–128.54) | 5.77 (122.26) | 5.77 (4.29) | 2.53 (0.86) |
| **Skin And Subcutaneous Tissue Disorders** | | | | | | |
| Angioedema | 20 | 39602 | 2.12 (1.37–3.29) | 4.95 (63.13) | 4.95 (3.43) | 2.31 (0.64) |

CI, confidence interval; N, number of cases of total AEs associated with MMI; ROR, reporting odds ratio; PRR, the proportional reporting ratio, EBGM, the empirical Bayes geometric mean; IC, Information Component; n, number of cases with suspected AEs associated with MMI.

y-axis represents the negative logarithm of the p-adjustment of the p-value after Fisher's exact test and Bonferroni correction. A positive y-axis indicates a highly significant difference. The color of the dots indicates the logarithm of the number of case reports. The more red the color, the higher the number of reports. Thus, the drugs in the upper right of the graph

(a)

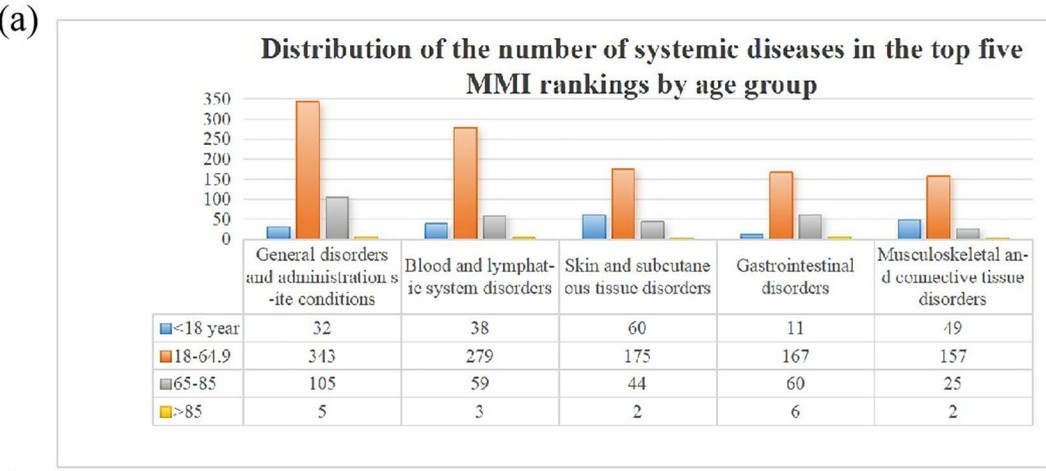

(b)

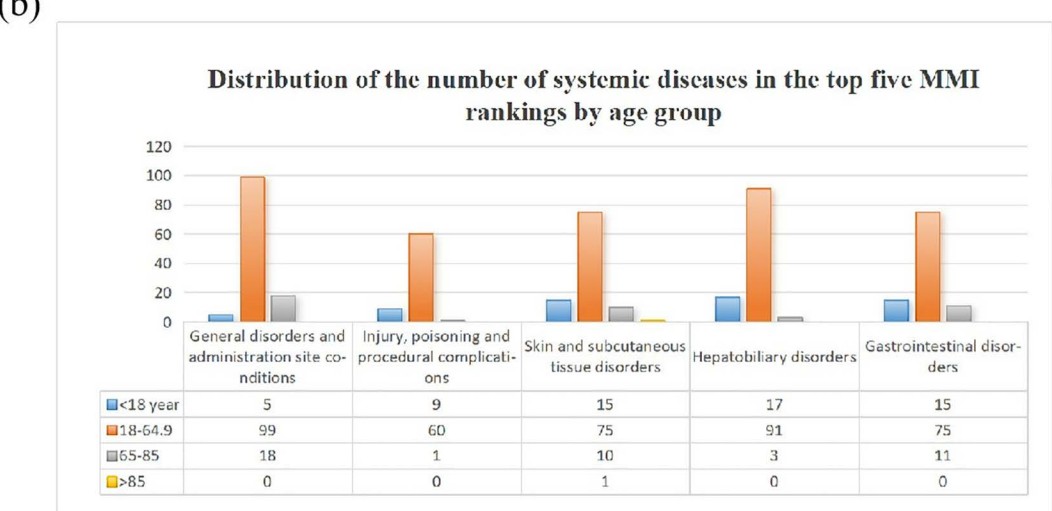

**Fig 3. Distribution of the number of systemic diseases in the top five methimazole and propylthiouracil rankings by age group.** MMI, methimazole; PTU, propylthiouracil.

have sinificant signal strength and differences. In order to better visualize the AEs of MMI/PTU, we also statistically produced the organ damage of MMI/PTU AEs based on SOC reports (Fig 5).

### 3.4. Time to onset of MMI/PTU-associated AEs

Of the 1586 patients with MMI, only 187 (11.8%) were available for time analysis. The median time to AEs in these patients was 31 days, with an interquartile range (IQR) of 31–74 days. As can be seen in Fig 6a, the majority of AEs in these patients (71.66%) occurred within the two months of MMI use. Meanwhile, of the 446 patients with PTU, only 40 (8.9%) were available for time analysis. The median time to AEs in these patients was 90 days, with an IQR of 20–388.5 days. As can be seen in Fig 6b, the majority of AEs in these patients (50.0%) occurred within the three months of PTU use. We also integrated MMI and PTU to analyze the time of SOC induction for skin and subcutaneous tissue disorders, hepatobiliary disorders, respiratory, thoracic and mediastinal disorders, blood and lymphatic system disorders and immune system disorders separately (Fig 6c–g). The majority of patients experienced skin and subcutaneous tissue disorders (60.2%) within the initial month, with a median time of 24 days and an IQR of 15–55 days; the

**Table 4. ROR with 95% CI for all positive PTU-related AEs.**

| AEs | PTU.n (n = 446) | Non.PTU.n (n = 52,157,560) | ROR 95%CI | PRR (χ2) | EBGM(EBGM05) | IC(IC025) |
|---|---|---|---|---|---|---|
| **Immune System Disorders** | | | | | | |
| Anti-Neutrophil Cytoplasmic Antibody Positive Vasculitis | 62 | 1617 | 1177.04 (909.32–1523.59) | 1135.64 (67692.32) | 1093.74 (881.32) | 10.1 (8.43) |
| **Injury, Poisoning And Procedural Complications** | | | | | | |
| Foetal Exposure During Pregnancy | 47 | 69389 | 20.58 (15.4–27.51) | 20.06 (851.79) | 20.05 (15.73) | 4.33 (2.66) |
| Exposure During Pregnancy | 30 | 84512 | 10.68 (7.44–15.32) | 10.51 (258.6) | 10.51 (7.77) | 3.39 (1.73) |
| **Blood And Lymphatic System Disorders** | | | | | | |
| Agranulocytosis | 28 | 14644 | 57.53 (39.59–83.6) | 56.63 (1527.68) | 56.53 (41.35) | 5.82 (4.15) |
| **Vascular Disorders** | | | | | | |
| Vasculitis | 23 | 9819 | 70.28 (46.56–106.1) | 69.38 (1546.68) | 69.22 (49.04) | 6.11 (4.45) |
| **Hepatobiliary Disorders** | | | | | | |
| Acute Hepatic Failure | 21 | 10975 | 57.34 (37.28–88.21) | 56.67 (1146.54) | 56.57 (39.45) | 5.82 (4.15) |
| Hepatic Failure | 14 | 26232 | 15.93 (9.41–26.95) | 15.81 (194.18) | 15.8 (10.17) | 3.98 (2.31) |
| Jaundice | 11 | 23989 | 13.66 (7.55–24.72) | 13.58 (128.2) | 13.58 (8.27) | 3.76 (2.1) |
| **Investigations** | | | | | | |
| Antineutrophil Cytoplasmic Antibody Positive | 19 | 506 | 1124.26 (709.43–1781.66) | 1112.14 (20329.6) | 1071.93 (729.21) | 10.07 (8.39) |
| Alanine Aminotransferase Increased | 11 | 53893 | 6.08 (3.36–10.99) | 6.05 (46.36) | 6.04 (3.68) | 2.6 (0.93) |
| Aspartate Aminotransferase Increased | 10 | 46629 | 6.38 (3.43–11.88) | 6.35 (45.12) | 6.35 (3.78) | 2.67 (1) |
| **Surgical And Medical Procedures** | | | | | | |
| Liver Transplant | 17 | 3043 | 167.07 (103.48–269.72) | 165.46 (2763.72) | 164.55 (110.21) | 7.36 (5.69) |
| **Respiratory, Thoracic And Mediastinal Disorders** | | | | | | |
| Respiratory Failure | 14 | 62856 | 6.64 (3.93–11.24) | 6.6 (66.54) | 6.6 (4.25) | 2.72 (1.05) |
| Pulmonary Alveolar Haemorrhage | 12 | 4354 | 82.18 (46.55–145.11) | 81.63 (953.16) | 81.41 (50.59) | 6.35 (4.68) |
| **Skin And Subcutaneous Tissue Disorders** | | | | | | |
| Cutaneous Vasculitis | 13 | 3868 | 100.28 (58.06–173.2) | 99.54 (1264.05) | 99.21 (62.8) | 6.63 (4.96) |
| **Nervous System Disorders** | | | | | | |
| Hepatic Encephalopathy | 12 | 8133 | 43.99 (24.93–77.65) | 43.7 (500.02) | 43.64 (27.13) | 5.45 (3.78) |
| **Renal And Urinary Disorders** | | | | | | |
| Glomerulonephritis Rapidly Progressive | 10 | 882 | 337.72 (180.76–630.98) | 335.81 (3300.73) | 332.05 (196.82) | 8.38 (6.71) |
| **General Disorders And Administration Site Conditions** | | | | | | |
| Multiple Organ Dysfunction Syndrome | 11 | 38099 | 8.6 (4.75–15.56) | 8.55 (73.38) | 8.55 (5.21) | 3.1 (1.43) |
| **Pregnancy, Puerperium And Perinatal Conditions** | | | | | | |
| Premature Baby | 11 | 26779 | 12.24 (6.76–22.14) | 12.17 (112.74) | 12.16 (7.41) | 3.6 (1.94) |
| **Congenital, Familial And Genetic Disorders** | | | | | | |
| Goitre Congenital | 10 | 25 | 11914.91 (5714.07–24844.84) | 11847.26 (84609.05) | 8462.61 (4575.77) | 13.05 (11.31) |

CI, confidence interval; N, number of cases of total AEs associated with PTU; ROR, reporting odds ratio; PRR, the proportional reporting ratio, EBGM, the empirical Bayes geometric mean; IC, Information Component; n, number of cases with suspected AEs associated with PTU.

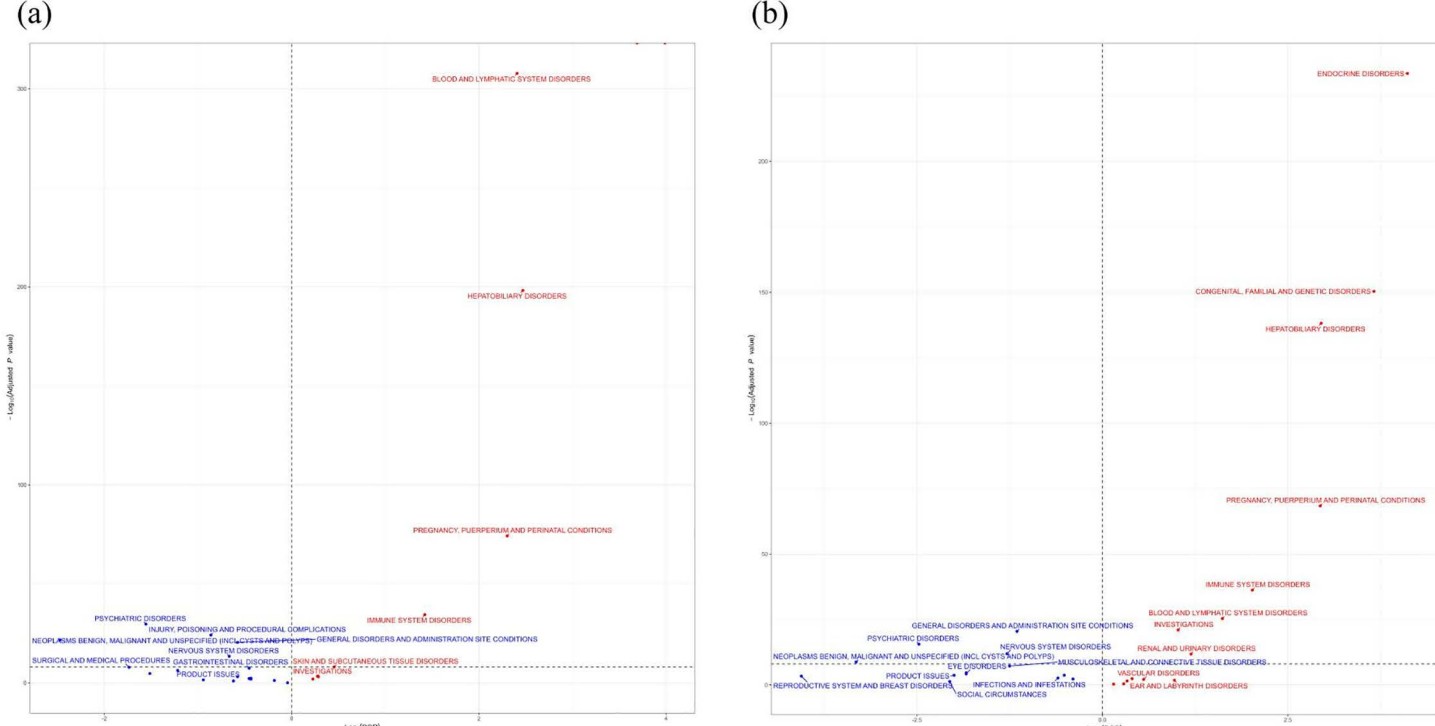

**Fig 4. SOC volcano diagrams associated with methimazole and propylthiouracil.** ROR, reporting odds ratio; P-adjust, p-value after Bonferroni correction.

majority of patients experienced hepatobiliary disorders (58.5%) within the two months, with a median time of 47 days and an IQR of 21.5–109 days; The majority of patients experienced respiratory, thoracic and mediastinal disorders within four months in 51.5% of patients and also 34.8% at a time greater than 360 days, with a median time of 92 days and an IQR of 58–143 days; the majority of patients experienced blood and lymphatic system disorders (62.7%) within two months, with a median time of 42 days and an IQR of 24–73.5 days; the majority of patients experienced immune system disorders within the initial month (20.0%) and within 181–360 days (22.2%), with a median time of 25.5 days and an IQR of 14–426 days.

## 4. Discussion

There is growing evidence that ATD is an effective treatment for thyrotoxicosis. [24–26] Particularly in Graves' disease which is the most common cause of thyrotoxicosis, ATD is the preferred approach for patients. [27] A cohort study of 1186 patients with Graves' disease with a follow-up of 10 years reported a high quality of life in patients treated with radioactive iodine compared to those receiving ATD. [28] MMI and PTU are the most commonly used ATDs. Therefore, it is essential to understand the AEs of MMI and PTU. Based on the FAERS database, this study presents an up-to-date safety profile of relevant AEs occurring in real populations for MMI and PTU from 2004 to 2024. To our knowledge, we were the first to conduct the most comprehensive study of AEs at MMI and PTU through the FAERS database. In our study, a total of 1584 patients who developed AEs after using MMI and 446 patients who developed AEs after using PTU were retained. In the 1584 patients with MMI and 446 patients with PTU, the AEs occurred more commonly in females (65.1% and 74.4%) than in males (18.9% and 15.5%), due to the pathogenetic features of thyrotoxicosis. [29] The global prevalence of Graves' disease is 2% in women and 0.5% in men. [1,30] Thyroiditis occurs in 8% of postpartum women. [31]

(a)

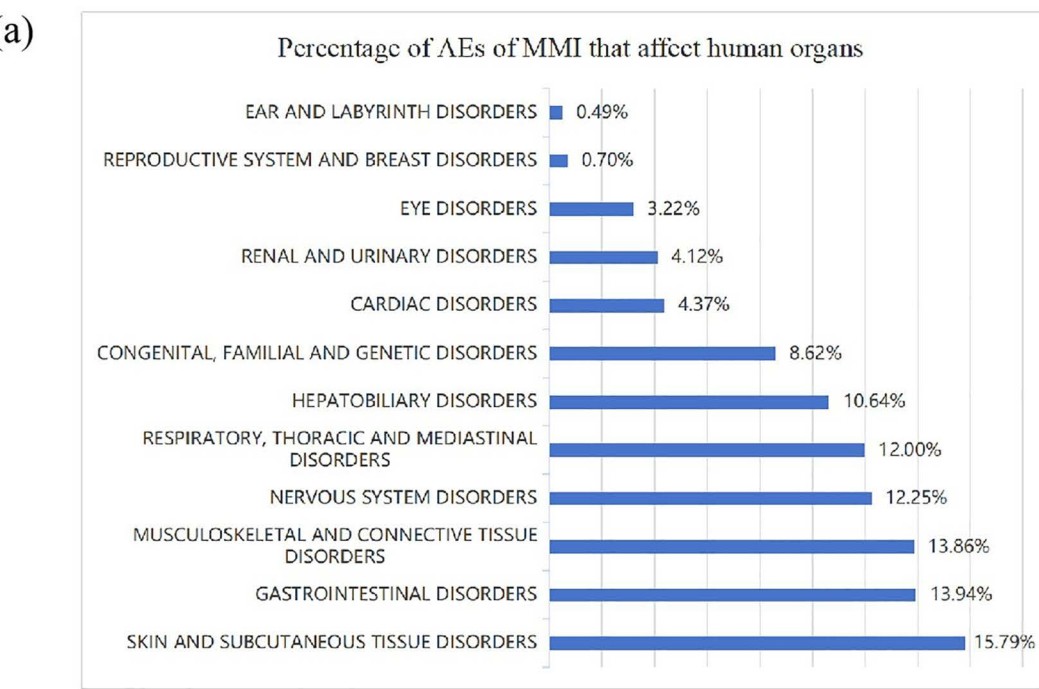

(b)

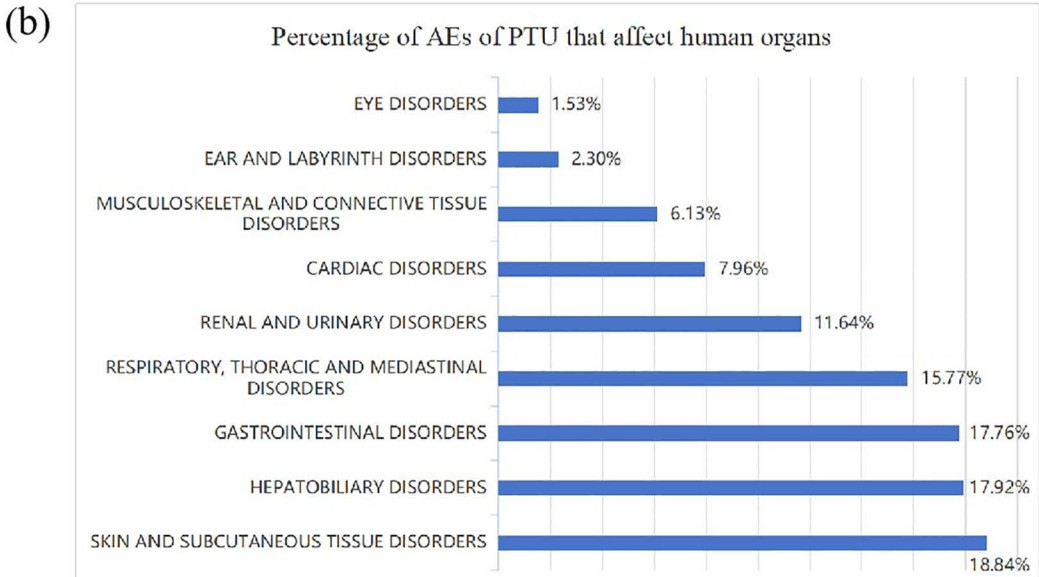

**Fig 5. Organ effects of AEs in methimazole and propylthiouracil.** MMI, methimazole; PTU, propylthiouracil.

Our results demonstrated that at the SOC level, the significant signals for both MMI and PTU were found for blood and lymphatic system disorders, skin and subcutaneous tissue disorders, respiratory, thoracic and mediastinal disorders, hepatobiliary disorders, congenital, familial and genetic disorders, endocrine disorders. The difference was that MMI had significantly disproportionate of AEs in musculoskeletal and connective tissue disorders, and PTU had significantly disproportionate of AEs in renal and urinary disorders, vascular disorders and ear and labyrinth disorders. It's worth noting

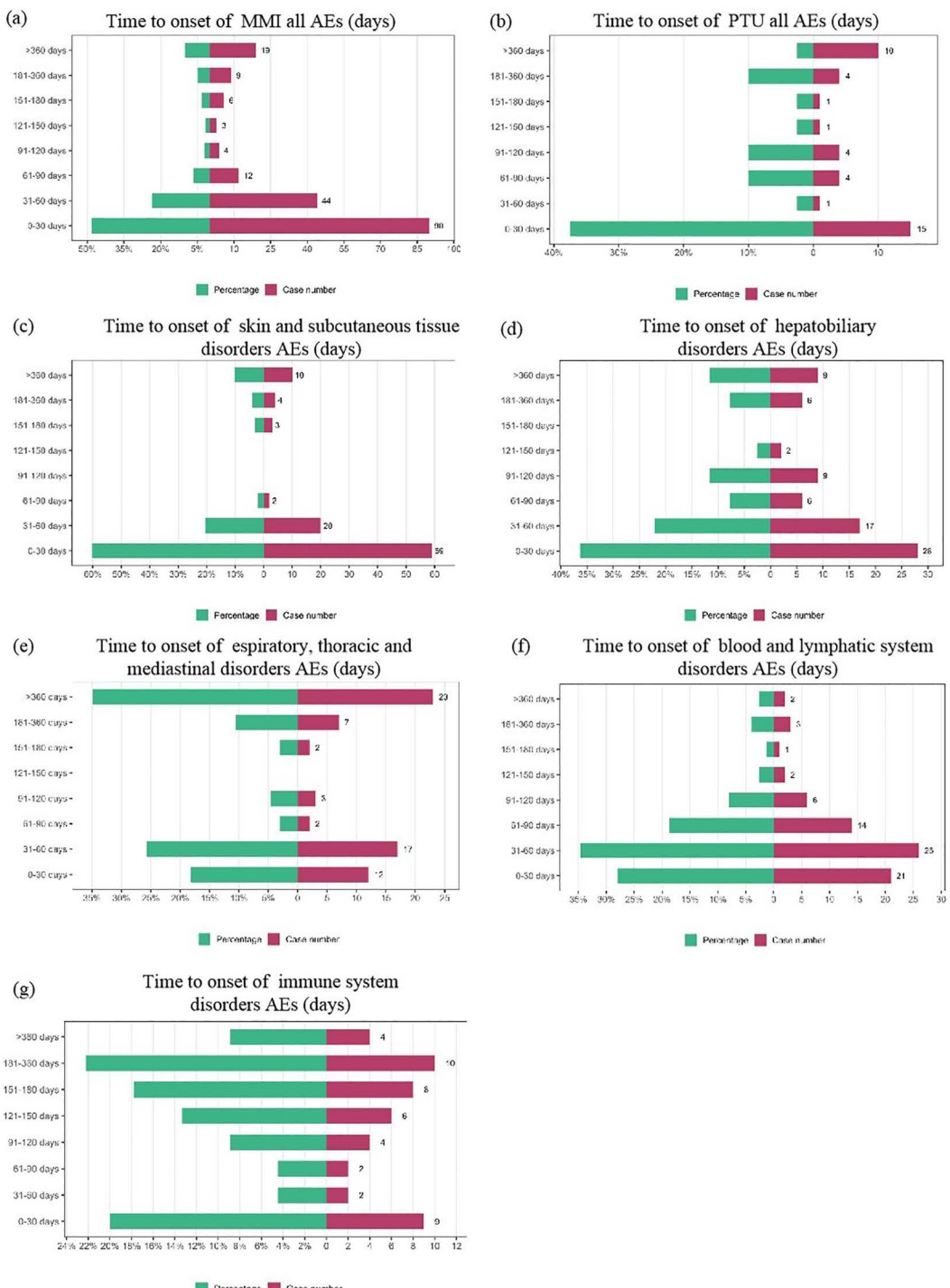

**Fig 6. Time to onset of adverse events of methimazole and propylthiouracil. (a)** Time to onset of all adverse effect of methimazole; **(b)** Time to onset of all adverse effect of propylthiouracil; **(c)** Time to onset of skin and subcutaneous tissue disorders of methimazole and propylthiouracil; **(d)** Time to onset of hepatobiliary disorders of methimazole and propylthiouracil; **(e)** Time to onset of espiratory, thoracic and mediastinal disorders of methimazole and propylthiouracil; **(f)** Time to onset of blood and lymphatic system disorders of methimazole and propylthiouracil; **(g)** Time to onset of immune system disorders of methimazole and propylthiouracil. MMI, methimazole; PTU, propylthiouracil; AEs, adverse events.

that four of the five most commonly AEs for MMI and PTU were the same, blood and lymphatic system disorders, skin and subcutaneous tissue disorders and respiratory, thoracic and mediastinal disorder, respectively.

The signals of disproportionality reporting in FAERS showed a high risk to blood and lymphatic system in MMI and PTU. Agranulocytosis was the most common AEs in blood and lymphatic system. Agranulocytosis is a life-threatening complication of ATDs that requires routine evaluation of the granulocyte count. Our result shows that the onset of agranulocytosis usually occurs 1–2 month, suggesting the necessity of routine monitoring within the first 2–3 months after ATDs initiation. Moreover, the immediate step to take after the onset of agranulocytosis is to stop the medication, after which the granulocyte count usually recovers. However, there is still controversy as to whether there is a difference in the risk of agranulocytosis with PTU and MMI. In 2020, a meta-analysis demonstrated that the agranulocytosis incidence did not significantly differ between the MMI and PTU groups. [32] However, in 2023, a real-world data study in Japan showed that the slightly higher risk with MMI than that with PTU in the first two weeks of administration. [33] In 2024, Yoshimura Noh et al. [34] found that the incidences of agranulocytosis induced by MMI and PTU exhibit dose-dependency and at comparable thyroid hormone synthesis inhibitory doses PTU has a considerably higher propensity to induce agranulocytosis than MMI does. In our study, the ROR of agranulocytosis for MMI use was approximately 2.4 times higher than that for PTU use. Because MMI-induced agranulocytosis and PTU-induced agranulocytosis, is dose dependent, and the risk of agranulocytosis for MMI and PTU use may also depend on a patient's genetic background, [35] the relative risk ratio between MMI and PTU use should be interpreted with caution.

For skin and subcutaneous tissue disorders, the most common symptom of PTU is cutaneous vasculitis, a serious adverse reaction during treatment. In our study, over 60% of skin and subcutaneous tissue disorders due to ATDs occurred in the first month after dosing, with a median of 24 days, which is in line with the results of a Danish multicenter study showing that skin reactions were the first to develop (median time: 30 days) and the most predominant AE of MMI. [36] PTU-induced cutaneous vasculitis, a manifestation of antineutrophil cytoplasmic antibody (ANCA)-associated vasculitis (AAV) involving the skin, has been reported previously. [37–39] Cutaneous vasculitis is a serious AE and can be fatal. Therefore, patients treated with PTU must be monitored frequently for such AEs. Discontinuation of the drug should be the first step in treatment and can lead to complete resolution of symptoms.

Hepatobiliary disorders may also result during MMI and PTU treatment. MMI-induced hepatobiliary disorders include drug-induced liver injury, jaundice, cholestasis, jaundice cholestatic, and hepatotoxicity. PTU-induced hepatobiliary disorders include acute hepatic failure, hepatic failure, and jaundice. Two previous meta-analyses showed that the risk of PTU-induced liver injury was higher than MMI. In our study, the ROR of acute for PTU use was approximately 2.4 times higher than that for PTU use. [32,40] In our study, the ROR of acute hepatic failure caused by the use of PTU (ROR = 57.34) was 5 times higher than that of MMI (ROR = 11.64), and the risk of hepatic failure caused by PTU (ROR = 15.93) was 2.6 times higher than that caused by MMI (ROR = 6), which is consistent with the results of previous studies. Over 58.5% of Hepatobiliary disorders due to ATDs occurred in the first two month after dosing. Therefore, patients' liver function needs to be monitored more carefully during the first two months after starting treatment with ATDs.

It's important to note that the immune system disorders associated with MMI and PTU, which the most serious is anti-neutrophil cytoplasmic antibody positive vasculitis. ANCA-positive vasculitis is a disorder characterized by severe, systemic, small-vessel vasculitis. The proportion of ANCA-positive cases was estimated at 4–64% in patients with PTU use and 0–16% in patients with MMI use [41]. In 2009, a post-marketing adverse event report showed that the estimated incidence of ANCA-associated vasculitis was 39.2 times higher with PTU than with MMI [42]. In 2022, In 2022, a study of Japanese databases found that the ROR of ANCA-associated vasculitis was 29.84-fold higher for PTU use than for MMI use [33]. However, the ROR of ANCA-associated vasculitis was only 7.59-fold higher for PTU use than for MMI use in our study. We suspect that this result may be due to the effect of racial differences.

It's worth noting that we found disproportionality reporting for respiratory, thoracic and mediastinal disorders. The respiratory, thoracic and mediastinal disorders for which MMI signals are significant and common are oropharyngeal pain and pleural

effusion, but the respiratory diseases for which PTU signals are significant and common are respiratory failure and pulmonary alveolar haemorrhage. These AEs have not been reported in the MMI and PTU instructions. MMI causing oropharyngeal pain has also not been reported in the literature, but there are reports of MMI causing pleural effusion. [43,44] PTU-induced respiratory failure and alveolar haemorrhage have both been reported, both of which may be secondary to vasculitis. [45–47]

Unexpected and significant safety signals such as polyarthritis, rhabdomyolysis, exomphalos, heart rate increased, angioedema, dysmorphism, and premature baby related to MMI were detected in our analysis. These AEs are not mentioned in the instructions for MMI. We also identified new PTU-associated PTs, such as antineutrophil cytoplasmic antibody positive, hepatic encephalopathy, multiple organ dysfunction syndrome, premature baby and goitre congenital, which are also AEs not documented in PTU instructions.

The results of the study showed that the median time to onset of MMI and PTU was 31 and 90 days, respectively, with the highest number of cases occurring in the first 1 month after drug therapy (48.1% vs 37.5%). These results suggest that we should be alert to AEs associated with MMI and PTU therapy during the first two month after treatment, and that AEs caused by MMI and PTU therapy should be recognized early, as these adverse drug reactions can be life-threatening.

While the current study shows a potentially profound relationship between PTU and MMI use and the odds of reporting AEs in FAERS, it is not without limitations. First, voluntary reporting is not inherently limited to healthcare professionals, and consumers can report AEs as well; unfortunately, however, consumers have limited medical expertise. Second, because the FDA does not require that reports be submitted to prove causation, some relevant AEs may be questionable. Further, the lack of information in the FAERS database makes it difficult to control for confounding factors such as age, comorbidities, or other factors that may affect health. However, while data mining techniques cannot compensate for the inherent limitations of spontaneous reporting systems or replace expert review, they do have their place, especially when large amounts of data are involved.

## Supporting information

**S1 File. Original data**.
(ZIP)

## Author contributions

**Conceptualization:** Qiong Sun.

**Data curation:** Hang Li.

**Funding acquisition:** Qin Long.

**Investigation:** Qiong Sun.

**Methodology:** Hang Li.

**Software:** Hang Li.

**Supervision:** Qin Long.

**Writing – original draft:** Yun Li.

**Writing – review & editing:** Yun Li.

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
