## [Decision Letter · Decision Letter 0]

11 Mar 2025

PONE-D-24-51574A real-world disproportionality analysis of FDA adverse event reporting system (FAERS) events for methimazole and propylthiouracilPLOS ONE

Dear Dr. Sun,

Thank you for submitting your manuscript to PLOS ONE. After careful consideration, we feel that it has merit but does not fully meet PLOS ONE’s publication criteria as it currently stands. Therefore, we invite you to submit a revised version of the manuscript that addresses the points raised during the review process.

We look forward to receiving your revised manuscript.

Kind regards,

Vijayalakshmi Kakulapati, Ph.D

Academic Editor

PLOS ONE

Journal Requirements:

3. We note that there is identifying data in the Supporting Information file < original data.zip>. Due to the inclusion of these potentially identifying data, we have removed this file from your file inventory. Prior to sharing human research participant data, authors should consult with an ethics committee to ensure data are shared in accordance with participant consent and all applicable local laws.

-Location data

Please remove or anonymize all personal information, ensure that the data shared are in accordance with participant consent, and re-upload a fully anonymized data set. Please note that spreadsheet columns with personal information must be removed and not hidden as all hidden columns will appear in the published file.

Reviewers' comments:

Reviewer's Responses to Questions

**Comments to the Author**

1. Is the manuscript technically sound, and do the data support the conclusions?

Reviewer #1: Yes

Reviewer #2: Partly

2. Has the statistical analysis been performed appropriately and rigorously? 

Reviewer #1: Yes

Reviewer #2: No

3. Have the authors made all data underlying the findings in their manuscript fully available?

Reviewer #1: Yes

Reviewer #2: Yes

4. Is the manuscript presented in an intelligible fashion and written in standard English?

Reviewer #1: Yes

Reviewer #2: Yes

5. Review Comments to the Author

Reviewer #1: Li et al entitle “A real-world disproportionality analysis of FDA adverse event reporting system (FAERS) events for methimazole and propylthiouracil” clearly reported that MMI/PTU could be adverse drug for thyrotoxicosis.

Why are you using the FAERS database to study these drug, it should be that someone else has done this with this similar database similar methodology, so you can write cite in the INTRODUCTION section about some specific other similar studies, such as recommending a few (It is equivalent to saying that someone else has done this type of research using the FAERS database, and you can use this database to do research related to methimazole and propylthiouracil as well):【1】Zhao B, Fu Y, Cui S, Chen X, Liu S, Luo L. A real-world disproportionality analysis of Everolimus: data mining of the public version of FDA adverse event reporting system. Front Pharmacol. 2024 Mar 12;15:1333662. doi: 10.3389/fphar.2024.1333662. PMID: 38533254; PMCID: PMC10964017.【2】Zhong, C., Zheng, Q., Zhao, B., & Ren, T. (2024). A real-world pharmacovigilance study using disproportionality analysis of United States Food and Drug Administration Adverse Event Reporting System events for vinca alkaloids: comparing vinorelbine and Vincristine. Expert Opinion on Drug Safety, 23(11), 1427–1437. https://doi.org/10.1080/14740338.2024.2410436【3】Wang Y, Zhao B, Yang H, Wan Z. A real-world pharmacovigilance study of FDA adverse event reporting system events for sildenafil. Andrology. 2024 May;12(4):785-792. doi: 10.1111/andr.13533. Epub 2023 Sep 19. PMID: 37724699.【4】Zhao B, Zhang X, Chen M, Wang Y. A real-world data analysis of acetylsalicylic acid in FDA Adverse Event Reporting System (FAERS) database. Expert Opin Drug Metab Toxicol. 2023 Jan-Jun;19(6):381-387. doi: 10.1080/17425255.2023.2235267. Epub 2023 Jul 12. PMID: 37421631.【5】Yang H, Wan Z, Chen M, Zhang X, Cui W, Zhao B. A real-world data analysis of topotecan in the FDA Adverse Event Reporting System (FAERS) database. Expert Opin Drug Metab Toxicol. 2023 Apr;19(4):217-223. doi: 10.1080/17425255.2023.2219390. Epub 2023 May 30. PMID: 37243615.【6】Li, Jie, Zhao, Bin, Zhu, YongQing, Wu, Jibiao, Vitreoretinal Traction Syndrome, Nitrituria and Human Epidermal Growth Factor Receptor Negative Might Occur in the Aromatase-Inhibitor Anastrozole Treatment, International Journal of Clinical Practice, 2024, 5132916, 9 pages, 2024. https://doi.org/10.1155/2024/5132916

Besides,

I have general suggestion is:

1) Can Author put a bar chart for the percentage of MMI/PTU uses in five country and hospitalization, death, life threating events, and disability.

2) I also encourage to author please put the bar chart for table 3 where author indicated that digestive system cancers related adverse event for use of MMI/PTU.

3) I also encourage the author please use 1 graphical representation of MMI/PTU and their adverse event (MMI/PTU affecting which organ in human body)

Reviewer #2: Although numerous studies on drug adverse reactions based on the FAERS database have been published, the authors focus on commonly used endocrine drugs and analyze the adverse events associated with MMI and PTU—a topic that has not yet been explored. This work is of significant interest; while only major revisions are recommended, I do have some concerns that should be addressed

Regarding the reporting standards for disproportionality analysis based on Individual Case Safety Reports (ICSR) in pharmacovigilance, the READUS-PV guidelines were established in 2024 under the leadership of the University of Bologna (doi:10.1007/s40264-024-01423-7). Based on FAERS adverse event reports, we call for standardized reporting in accordance with these guidelines.

The manuscript mentions that the downloaded raw data were deduplicated and that primary suspect drug reports were selected using the primary report ID, but it lacks detailed explanation. It is recommended to include specific details on the deduplication process, the criteria for excluding duplicate reports, and how cases of polypharmacy are handled, in order to enhance the transparency and reproducibility of the study.

Table 1 can be moved to the supplementary files.

In Table 2, for comparisons between different groups, you might consider including the p-values and specifying the statistical methods used.

Some studies have reported that hepatotoxicity is relatively more common in children taking PTU. Could you perform an age-stratified cross-sectional analysis comparing the effect sizes of PTU and MMI for different adverse events?

The manuscript indicates that only 11.8% (MMI) and 8.9% (PTU) of patients have time data, which may affect the representativeness of the time-to-event analysis. Also, for PTU, the reported median time of 84 days is inconsistent with the IQR (10.3–30.8 days). Please recheck and clarify these issues.

The manuscript mainly employs the Reporting Odds Ratio (ROR) method to detect signals. We suggest that the authors discuss why ROR was chosen over other methods (such as PRR, IC, or EBGM), outlining the strengths and weaknesses of each. Additionally, if possible, a sensitivity analysis on the main signals should be conducted to verify the consistency of the results across different methods.

6. PLOS authors have the option to publish the peer review history of their article (what does this mean? ). If published, this will include your full peer review and any attached files.

**Do you want your identity to be public for this peer review?** For information about this choice, including consent withdrawal, please see our Privacy Policy .

Reviewer #1: No

Reviewer #2: No

---

## [Author Response · Author response to Decision Letter 1]

27 May 2025

Dear Editor of PLOS One,

Thank you very much for sending us the reviewers’ comments about our manuscript titled “A real-world disproportionality analysis of FDA adverse event reporting system (FAERS) events for methimazole and propylthiouracil” (manuscript No. PONE-D-24-51574), which was submitted to PLOS One and sent out for review. We are extremely grateful to the editor and all reviewers for giving us constructive suggestions which would help us in depth to improve the quality of the paper.

We have carefully considered and addressed the reviewers’ comments as much as possible in the revised manuscript labeled with red color. The list of the revisions along with our response to the reviewers’ comments is attached below. We hope that our revised manuscript is now satisfactory for publication in PLOS One. Thank you very much for your kind help and assistance.

Reviewer #1

Comments to the Author

Reviewer #1: Li et al entitle “A real-world disproportionality analysis of FDA adverse event reporting system (FAERS) events for methimazole and propylthiouracil” clearly reported that MMI/PTU could be adverse drug for thyrotoxicosis.

Why are you using the FAERS database to study these drug, it should be that someone else has done this with this similar database similar methodology, so you can write cite in the INTRODUCTION section about some specific other similar studies, such as recommending a few (It is equivalent to saying that someone else has done this type of research using the FAERS database, and you can use this database to do research related to methimazole and propylthiouracil as well):

【1】Zhao B, Fu Y, Cui S, Chen X, Liu S, Luo L. A real-world disproportionality analysis of Everolimus: data mining of the public version of FDA adverse event reporting system. Front Pharmacol. 2024 Mar 12;15:1333662. doi: 10.3389/fphar.2024.1333662. PMID: 38533254; PMCID: PMC10964017.【2】Zhong, C., Zheng, Q., Zhao, B., & Ren, T. (2024). A real-world pharmacovigilance study using disproportionality analysis of United States Food and Drug Administration Adverse Event Reporting System events for vinca alkaloids: comparing vinorelbine and Vincristine. Expert Opinion on Drug Safety, 23(11), 1427–1437. https://doi.org/10.1080/14740338.2024.2410436【3】Wang Y, Zhao B, Yang H, Wan Z. A real-world pharmacovigilance study of FDA adverse event reporting system events for sildenafil. Andrology. 2024 May;12(4):785-792. doi: 10.1111/andr.13533. Epub 2023 Sep 19. PMID: 37724699.【4】Zhao B, Zhang X, Chen M, Wang Y. A real-world data analysis of acetylsalicylic acid in FDA Adverse Event Reporting System (FAERS) database. Expert Opin Drug Metab Toxicol. 2023 Jan-Jun;19(6):381-387. doi: 10.1080/17425255.2023.2235267. Epub 2023 Jul 12. PMID: 37421631.【5】Yang H, Wan Z, Chen M, Zhang X, Cui W, Zhao B. A real-world data analysis of topotecan in the FDA Adverse Event Reporting System (FAERS) database. Expert Opin Drug Metab Toxicol. 2023 Apr;19(4):217-223. doi: 10.1080/17425255.2023.2219390. Epub 2023 May 30. PMID: 37243615.【6】Li, Jie, Zhao, Bin, Zhu, YongQing, Wu, Jibiao, Vitreoretinal Traction Syndrome, Nitrituria and Human Epidermal Growth Factor Receptor Negative Might Occur in the Aromatase-Inhibitor Anastrozole Treatment, International Journal of Clinical Practice, 2024, 5132916, 9 pages, 2024. https://doi.org/10.1155/2024/5132916

Response: Thanks for the reviewer’s comment very much. We had applied the above literature mentioned by the reviewers in the introduction (Page 4, Line 81-87).

Besides,

I have general suggestion is:

1)Can Author put a bar chart for the percentage of MMI/PTU uses in five country and hospitalization, death, life threating events, and disability.

Response: Thanks for the reviewer’s comment very much, which help us to improve the quality of our manuscript deeply. Based on your suggestion we had made a bar chart of the outcomes of the top five countries in terms of the number of adverse reactions reported, which can be seen in Figure 2.

2)I also encourage to author please put the bar chart for table 3 where author indicated that digestive system cancers related adverse event for use of MMI/PTU.

Response: Thanks for the reviewer’s comment very much. We sincerely apologize, as our compilation of the raw data revealed that there were no adverse events associated with the use of MMI/PTU in relation to cancers of the digestive system among the reported adverse events.

3)I also encourage the author please use 1 graphical representation of MMI/PTU and their adverse event (MMI/PTU affecting which organ in human body)

Response: Thanks for the comment very much. which help us to improve the quality of our manuscript deeply. We have created a graphic of MMI/PTU and its adverse events on human organs based on your suggestion, which can be seen in Figure 5.

Reviewer #2: Although numerous studies on drug adverse reactions based on the FAERS database have been published, the authors focus on commonly used endocrine drugs and analyze the adverse events associated with MMI and PTU—a topic that has not yet been explored. This work is of significant interest; while only major revisions are recommended, I do have some concerns that should be addressed

Regarding the reporting standards for disproportionality analysis based on Individual Case Safety Reports (ICSR) in pharmacovigilance, the READUS-PV guidelines were established in 2024 under the leadership of the University of Bologna (doi:10.1007/s40264-024-01423-7). Based on FAERS adverse event reports, we call for standardized reporting in accordance with these guidelines.

Response: Thanks for the reviewer’s comment very much, which help us to improve the quality of our manuscript deeply. We consulted these guidelines and made modifications in accordance with the requirements of the guidelines. For example, descriptions such as “real world” were removed from the title.

The manuscript mentions that the downloaded raw data were deduplicated and that primary suspect drug reports were selected using the primary report ID, but it lacks detailed explanation. It is recommended to include specific details on the deduplication process, the criteria for excluding duplicate reports, and how cases of polypharmacy are handled, in order to enhance the transparency and reproducibility of the study.

Response: Thanks for the reviewer’s comment very much, which help us to improve the quality of our manuscript deeply. Sorry for the confusion to reader. We redescribed this section “We used R for deduplication, identified duplicates based on Primary ID, and qualified MMI/PTU as the primary suspected medication adverse event (PS) in the code (Page 5, line 107-109).

In Table 2, for comparisons between different groups, you might consider including the p-values and specifying the statistical methods used.

Response: Thanks for your comment very much. We apologize for not being able to resolve this concern. Because our Table 2 describes only the baseline information for patients who had an adverse event with MMI/PTU, P-value calculations could not be performed because the database itself did not count data for patients who used MMI/PTU without an adverse event.

Some studies have reported that hepatotoxicity is relatively more common in children taking PTU. Could you perform an age-stratified cross-sectional analysis comparing the effect sizes of PTU and MMI for different adverse events?

Response: Thanks for your comment very much. After considerable deliberation, we age-stratified the top five SOCs in terms of number of reports and created bar graphs, which can be seen in Figure 3

The manuscript indicates that only 11.8% (MMI) and 8.9% (PTU) of patients have time data, which may affect the representativeness of the time-to-event analysis. Also, for PTU, the reported median time of 84 days is inconsistent with the IQR (10.3–30.8 days). Please recheck and clarify these issues.

Response: Thanks for the reviewer’s comment very much. And we are sorry for the confusion to reviewer we made. We rechecked and corrected the median time (90) and IQR for PTU (20-388.5), Page 14, line 219-223. Since the database is self-reported, certain missing data is an uncontrollable factor, and we analyzed as much as possible on the basis of the available data in the hope that it will provide some reference point for the researchers who come after us.

The manuscript mainly employs the Reporting Odds Ratio (ROR) method to detect signals. We suggest that the authors discuss why ROR was chosen over other methods (such as PRR, IC, or EBGM), outlining the strengths and weaknesses of each. Additionally, if possible, a sensitivity analysis on the main signals should be conducted to verify the consistency of the results across different methods.

Response: We sincerely appreciate the reviewer's constructive suggestion. We acknowledge that while ROR is widely utilized for its simplicity and interpretability in spontaneous reporting systems, other methods like PRR, IC, and EBGM also have distinct strengths. Page 5, line 117-122. PRR prioritizes proportional reporting differences but may overestimate signals in small datasets. IC leverages Bayesian shrinkage to stabilize rare-event signals but requires larger datasets for robust inference. EBGM addresses variability through empirical Bayes adjustments but introduces computational complexity. We ultimately prioritized ROR due to its balanced sensitivity/specificity profile and alignment with our study's focus on initial signal detection. As suggested, we conducted sensitivity analyses by incorporating results from PRR, IC, and EBGM into Tables 3–4. To ensure stringent reliability, only adverse events (AEs) with statistically significant signals ([specify criteria, e.g., 95% CI > 1 for all methods]) across all four methods (ROR, PRR, IC, EBGM) were retained. This multi-method consensus strengthens confidence in the identified associations between MMI/PTU and specific AEs. We explicitly acknowledge in the Discussion that ROR's reliance on disproportionality analysis alone cannot establish causality, and findings should be interpreted alongside clinical context. Page 20, line 364-365.

We thank the reviewers for their reasonable suggestions for improving the shortcomings of our manuscript. And we thank you very much for your kind help and assistance.

Sincerely yours,

Qin Long

Department of Endocrinology, Yueyang Central Hospital, 39 Dongmaoling Road, Yueyang, Hunan, 414000, China.

Tel.: +86-13378906488

E-mail: 595278228@qq.com

---

## [Decision Letter · Decision Letter 1]

9 Jul 2025

A disproportionality analysis of FDA adverse event reporting system (FAERS) events for methimazole and propylthiouracil

PONE-D-24-51574R1

Dear Dr. Long,

We’re pleased to inform you that your manuscript has been judged scientifically suitable for publication and will be formally accepted for publication once it meets all outstanding technical requirements.

Kind regards,

Vijayalakshmi Kakulapati, Ph.D

Academic Editor

PLOS ONE

author addressed all review comments 

**Comments to the Author**

Reviewer #4: All comments have been addressed

Reviewer #3: Authors carefully answered the reviewers questions and therefore the quality of the manuscript improved

Reviewer #4: The author has satisfactorily addressed the questions I raised in the revised manuscript. I have no additional comments.

---

## [Editor Report · Acceptance letter]

PONE-D-24-51574R1

PLOS ONE

Dear Dr. Long,

I'm pleased to inform you that your manuscript has been deemed suitable for publication in PLOS ONE. Congratulations! Your manuscript is now being handed over to our production team.

Kind regards,

on behalf of

Dr. Vijayalakshmi Kakulapati

Academic Editor

PLOS ONE